# The Effects of Two Weeks of Oral PeakATP^®^ Supplementation on Performance during a Three-Minute All out Test

**DOI:** 10.3390/jfmk8020042

**Published:** 2023-04-04

**Authors:** Trevor J. Dufner, Jessica M. Moon, David H. Fukuda, Adam J. Wells

**Affiliations:** School of Kinesiology and Rehabilitation Sciences, University of Central Florida, 12494 University Blvd, Orlando, FL 32816, USA

**Keywords:** ATP, 3MT, power, exercise, PeakATP

## Abstract

Exogenous ATP has been shown to increase total weight lifted during resistance training interventions and attenuate fatigue during repeated Wingate assessments. However, the influence of exogenous ATP on single bout maximal effort performance has yet to be examined. The purpose of this study was to investigate the effects of PeakATP^®^ supplementation on performance during a 3-min all-out test (3MT). Twenty adults (22.3 ± 4.4 years, 169.9 ± 9.5 cm, 78.7 ± 14.6 kg) completed two identical 3MT protocols in a double-blind, counter-balanced, crossover design. Participants were randomized to either PeakATP^®^ (400 mg·day^−1^) or placebo (PLA) treatments and consumed their assigned supplement for 14 days and ingested an acute dose 30 min before each 3MT. A 14-day wash-out period was completed between each supplementation period and subsequent 3MT. Peak power, time to peak power, work above end power, end power, and fatigue index were assessed during each 3MT. Dependent *t*-tests and Hedge’s *g* effect sizes were used to assess differences between treatments. No significant differences were observed between treatments for 3MT performance (*p* > 0.05). These findings indicate that 3MT performance was not significantly impacted by PeakATP^®^ supplementation. This may be due in part to the continuous nature of the 3MT as disodium ATP has been shown to be beneficial for repeated bout activities.

## 1. Introduction

Adenosine 5′triphosphate (ATP), commonly known as the “energy currency” of the body, is required for all active biological processes within the cell. ATP’s intracellular effects are well known and have long been recognized. However, the effects of exogenous ATP have only recently been investigated.

Extracellular concentrations of ATP (10–100 nM) remain relatively low when compared with intracellular ATP due to its short half-life and tight regulation by soluble enzymes of the ectonucleoside family [1,2,3,4]. Once metabolized, exogenous ATP is rapidly taken up and stored by erythrocytes which are thought to play a crucial role in maintaining plasma concentrations of extracellular ATP [2]. Due to erythrocyte’s ability to “sense” oxygen depleted tissues [5], this creates an effective transportation model wherein erythrocytes absorb, transport, and distribute metabolized ATP to tissues during periods of acute hypoxia, which may be seen in the muscle during high-intensity exercise. At the tissue, ATP is thought to bind to the P2Y receptor on the endothelium and stimulate a vasodilative response, resulting in increased blood flow to the tissue along with the potential for enhanced nutrient delivery and metabolite removal [6,7,8]. Metabolites of ATP such as adenosine may also stimulate purinergic receptors and cause hyperpolarization and vasodilation of arterioles, further enhancing ATP’s hemodynamic effects [9,10,11]. Furthermore, extracellular ATP has been shown to stimulate the P2X4 receptor in vitro, leading to greater intracellular calcium influx and enhanced muscle contractility [12,13]. Accordingly, exogenous ATP has the potential to mitigate fatigue during exercise by promoting substrate and calcium availability and removing metabolic waste from the working muscle.

Investigations examining orally administered ATP have yielded somewhat mixed results. For example, Jordan et al. [14] report no beneficial effect of 125 mg or 250 mg disodium ATP on Wingate performance acutely or after 14 days of supplementation. However, others have reported beneficial effects in response to larger daily doses of 400 mg. An acute dose of 400 mg of ATP 30 min prior to testing has been found to increase total weight lifted during 4 sets of half-squats at 80% 1 RM completed to momentary failure [15]. Two hundred mg twice daily ATP for 15 days has been shown to attenuate declines in low peak torque during set 2 of a 3 × 50 knee extension protocol [16], while 400 mg once daily ATP for 14 days has been shown to attenuate declines in muscle excitability in bouts 8–10 and improve peak power in bouts 8 and 10 of a 10 × 30 s repeated Wingate protocol [6]. Collectively, these findings suggest that acute and long-term supplementation with 400 mg·day^−1^ oral ATP may enhance exercise performance during repeated bout type activities, which is consistent with the proposed mechanisms of action. Notwithstanding, the effect of ATP supplementation on performance in a single bout of continuous high-intensity exercise is unknown. Understanding the effects of ATP supplementation on single bout exercise may be beneficial for those looking to improve single repetition and highly anaerobic single bout performances, potentially giving them a competitive edge in training and sport. Therefore, the purpose of this study was to examine the effects of once daily PeakATP^®^ supplementation (400 mg·day^−1^) for 14 days in conjunction with an acute dose 30 min prior to exercise on anaerobic and aerobic power performance during a 3-min all-out test (3MT). We hypothesized that once daily PeakATP supplementation for 14 days in combination with an acute dose 30 min prior to a 3MT would improve anaerobic work and end power performance.

## 2. Materials and Methods

### 2.1. Study Design

This study followed a randomized double-blind, placebo controlled, cross-over design. Data from each participant were collected over 5 visits to the research facility. The initial visit consisted of informed consent, a physical activity readiness questionnaire (PAR-Q+), and a medical and activity history questionnaire (MHQ) to determine eligibility for the study. During visit 2, participants underwent anthropometric assessments consisting of height, weight, and body composition via bioelectric impedance analysis (BIA). During visit 3, participants completed a standardized warm-up before performing a Maximal Aerobic Power (MAP) test on a cycle ergometer. Power output at gas exchange threshold (GET) and peak power output (PPO) were recorded during the MAP test. Following visit 3, participants were randomized to either supplement (PeakATP^®^) or placebo (PLA) and were instructed to consume their assigned supplement for 14 days. Following supplementation, participants returned to the research facility within 24 h of their final dose for the first of two experimental trials (T1, visit 4). During T1, participants ingested an acute dose of their assigned supplement 30 min before completing a standardized warm-up and 3MT. Following completion of T1, participants underwent a 14-day wash-out period followed by 14 days of supplementation with the alternate supplement (PeakATP or PLA). Participants returned to the research facility 24 h following their last dose to complete experimental trial 2 (T2, visit 5). T2 occurred in an identical fashion to T1 with participants consuming an acute dose of their assigned supplement 30 min prior to the exercise testing. The study protocol and design were approved by the University of Central Florida Institutional Review Board for Human Participants. The study was carried out in Orlando, Florida from August 2021 to May 2022. This study was registered with clinicaltrials.gov under the identifier NCT05100589.

### 2.2. Participants

A convenience sample of 20 healthy, recreationally active individuals (10 men and 10 women) from the University of Central Florida completed this study (age 22.3 ± 4.4 years, weight 78.7 ± 14.6 kg, height 170.0 ± 9.5 cm, lean mass 58.4 ± 14.2 kg, and body fat percentage 27.0 ± 9.5%). The sample size for this study was determined based on an a priori power analysis for changes in performance during a Go/No-Go response inhibition task following high-intensity exercise that was addressed as part of a separate research question examining the effects of the 3MT on cognition. To be included in this study, participants were required to be between the ages of 18 and 40, healthy, ready for activity as determined by a PAR-Q+ and MHQ, classified as recreationally active (>150 min of exercise per week), and not taking and willing to abstain from taking creatine or beta-alanine supplementation or willing to complete a 4-week wash-out prior to enrolling. Additionally, participants were instructed to refrain from caffeine consumption within 24 h of each trial, which was verbally confirmed upon their arrival to the testing location.

### 2.3. Anthropometrics

Anthropometry and body composition assessments took place during visit 2. Participants were required to arrive at the research facility at least two hours fasted and were asked to void their bladder and empty their pockets prior to the completion of the height, weight, and body composition assessments. Height and weight were assessed using a stadiometer and scale (Health-o-meter Professional Patient Weighing Scale, Model 500 KL, Pelstar, Alsip, IL, USA). Body composition was assessed via BIA (InBody 770, Biospace Co, Ltd. Seoul, Republic of Korea). Prior to BIA testing, participants were asked to remove shoes, socks, and all jewelry.

### 2.4. Maximal Aerobic Power (MAP) Test

During visit 3, participants performed a ramp protocol to volitional exhaustion on a cycle ergometer (Lode, Excalibur Sport, Groningen, the Netherlands). Prior to the test, participants completed a warm-up consisting of 5 min of light cycling at an intensity of 50 watts at a self-selected pace, 10 body weight squats, 10 body weight walking lunges, 10 dynamic waking hamstring stretches, and 10 dynamic walking quadricep stretches. The MAP protocol required each participant to maintain a pedaling cadence of 70–80 revolutions per minute (RPM) at an initial workload of 100 watts (W). Seat height was recorded for all participants and standardized for subsequent experimental trials. The workload increased 30 W every two minutes (1 W per 2 s) until participants were unable to maintain a cadence above 70 RPM for ~10 s despite verbal encouragement or volitional fatigue. Expired gasses were analyzed using open-circuit spirometry (True One 2400^®^ Metabolic Measurement System, Parvo-Medics Inc., Sandy, UT, USA). The highest power output achieved was recorded as PPO in watts (W). GET was determined via computerized regression analysis of the slopes of the CO_2_ uptake (VCO_2_) vs. the O_2_ uptake (VO_2_). Power at the GET was recorded.

### 2.5. Three-Minute All-Out Assessment (3MT)

The 3MT test was completed on a cycle ergometer (Lode, Excalibur Sport, Groningen, the Netherlands). Participants completed a standardized warm up that was identical to the one performed during the MAP test. Resistance during the 3MT was set as a function of the pedaling rate using a scaling factor that was based on the power output at a set cadence of 80 RPM being equal to 50% of the difference between the power output at GET and PPO assessed during the MAP test [17,18]. The test began with the participants completing the preparation phase on the cycle ergometer where they pedaled at 70–80 RPM for 1 min at a set resistance of 50 watts. Over the last 5 s of the preparation phase, participants were instructed to begin pedaling as maximally as possible. Participants were blinded to the total elapsed time and instructed to not pace themselves by giving maximal effort throughout the entire testing period. Verbal encouragement was given throughout the entire 3-min testing period. Peak power (PP), time to peak power (TPP), work above end power (WEP), end power (EP), and fatigue index (FI) were recorded during the 3MT for later analysis. EP was defined as the average power output over the final 30 s of the 3MT, while WEP was defined as the power–time integral above EP, and FI was defined as the percentage drop from the peak to the lowest power output.

### 2.6. Supplementation

Participants were assigned to consume either PeakATP or PLA for a 14-day period prior to the completion of each experimental trial in a randomized, double-blind, cross-over fashion. Treatment assignments were matched by sex across the two conditions. Participants also ingested an acute dose of the assigned supplement upon arrival to the lab 30 min prior to completing the 3MT. PeakATP (400 mg adenosine 5′-triphosphate disodium, maltodextrin, silica-colloidal anhydrous, citric acid anhydrous, sucralose, and guar gum) and PLA (maltodextrin, silica-colloidal anhydrous, citric acid anhydrous, sucralose, and guar gum) were obtained from TSI Group Ltd. (Missoula, MT, USA). Both PeakATP and PLA were provided in pre-portioned single serve stick packs in the form of a flavored powder that were similar in taste and appearance. Each participant was provided with a 14-day supply of their assigned formula following visit 3 and T1 (visit 4). A 14-day wash-out period was required following the completion of T1. Participants were instructed to mix their assigned formula in 8 oz of water and take 30 min before breakfast on an empty stomach. Participants were required to keep a daily log detailing the date and time each dose was ingested as well as any perceived side effects associated with the supplement. Participants were required to return all empty packets before the beginning of each experimental trial. Any remaining supplement was counted and recorded. During each experimental trial, the supplement was taken immediately upon arrival to the lab 30 min prior to the completion of the 3MT. Supplements were coded, assigned, and administered in a double-blind fashion. A sealed envelope containing the identity of each coded supplement was provided by the supplement manufacturer prior to the beginning of the study. Blinding was maintained throughout data collection and analysis and was unblinded upon completion of the statistical analysis.

### 2.7. Statistical Analysis

Paired sample *t*-tests were used to examine mean differences between treatments (PeakATP vs. PLA) in PP, TPP, WEP, EP, and FI during the 3MT. Data normality was assessed using the Shapiro–Wilk test for each treatment independently. Differences between treatments were further analyzed using Hedge’s g effect sizes. Hedge’s g results were interpreted using thresholds of <0.2, 0.2 to <0.6, 0.6 to <1.2, 1.2 to <2.0, and 2.0 to 4.0, which correspond to trivial, small, moderate, large, and very large ES, respectively. Since estimates for g may show positive bias with small sample sizes, a correction was applied to provide a more accurate estimate of the effect size (Equation (1)) [19]. Data were analyzed using SPSS statistical software (v. 28.0.1.0, SPSS Inc., Chicago, IL, USA). Significance for all analyses was accepted at an alpha level of *p* = 0.05.
(1)$$g=\frac{x_{2}-x_{1}}{\sqrt{\frac{\left(n_{1}-1\right)S_{1}^{2}+\left(n_{2}-1\right)S_{2}^{2}}{\left(n_{1}-1)+(n_{2}-1\right)}}}\times 1-\frac{3}{4n-9}$$
where *n* = the number of observations for each time point/treatment and *S* = the SD of the observations.

## 3. Results

### 3.1. Participant Compliance

All participants were found to be in compliance with the supplementation protocol. Compliance for the PeakATP group was 98.6% and 96.9% for the PLA group. No adverse side effects were reported for either treatment.

### 3.2. Demographics

Participant characteristics are shown in Table 1. Thirty-five participants were enrolled in this study. Of those thirty-five participants, eleven dropped out due to personal reasons, three were excluded due to their prescription medications, and one was excluded from analysis due to unusable data. Therefore, a final sample size of *n* = 20 was used for the analysis.

### 3.3. PeakATP Effects on Exercise Outcomes

No significant differences were observed for PP, TPP, WEP, EP, or FI between PeakATP and PLA (*p* > 0.05) (Table 2.)

## 4. Discussion

The purpose of this study was to compare the effects of oral supplementation with 400 mg disodium ATP versus PLA on PP, TPP, WEP, EP, and FI during a 3MT. It was hypothesized that 2 weeks of once daily PeakATP supplementation in combination with an acute dose 30 min prior to exercise would improve PP, TPP, WEP, EP, and attenuate fatigue. The results of this study, however, indicate no significant effects of oral supplementation with PeakATP on 3MT performance when compared with PLA, while the effect sizes for the performance variables were trivial to small.

The findings of this study are consistent with other studies that have shown no beneficial effect of exogenous ATP on anaerobic performance. Jordan et al. [14] examined the effects of low dose (150 mg) and high dose (225 mg) ATP supplementation on single Wingate performance both acutely (75 min after ATP ingestion) and following 14 days of supplementation in 27 recreationally active men who were undergoing strength training. They found no between group differences from baseline for peak power, amount of work performed, or average power at either time point when compared with the placebo. In another study, Rathmacher et al. [16] reported no effect of once daily supplementation with 400 mg ATP for 15 days on high peak torque, power or total work during a 3 × 50 maximal knee extension protocol, although an acute does of ATP was not provided prior to exercise.

In contrast to our findings, a number of studies indicate that 400 mg ATP may be efficacious in improving anaerobic performance. Freitas et al. [15] reported that a single 400 mg dose of ATP was sufficient to increase the total weight lifted and the number of repetitions completed during a single resistance training session in resistance trained individuals. Purpura et al. [6] reported that supplementation with 400 mg·day^−1^ ATP for 14 days attenuated the decline in muscle excitability in bouts 8–10 and improved peak power in bouts 8 and 10 of a 10 × 30 s repeated Wingate protocol. Lastly, Rathmacher et al. [16] reported greater low peak torque during set two of a 3 × 50 maximal knee extension protocol and a trend toward decreased torque fatigue during set three following 15 days of once daily supplementation with 400 mg ATP versus a placebo. Discrepancies in findings may be related to differences in the exercise protocols that the studies utilized. For example, extensive localized tissue ischemia has been demonstrated during continuous high-intensity exercise, with mean skeletal muscle tissue oxygenation of the vastus lateralis declining from 74.4% at rest to 36.4% at maximal oxygen consumption during a cycle based ramp protocol to exhaustion [20]. Notably, tissue oxygenation was shown to rapidly return to above baseline levels (82.3%) 60 s post-exercise, indicating that the provision of recovery between bouts of high-intensity exercise is critical for the reperfusion of skeletal muscle. ATP supplementation has been previously shown to influence hemodynamics at rest and post-exercise in conjunction with repeated bout activities by promoting nitric oxide release and subsequent vasodilation [21,22,23]. This may allow for increased blood flow to the working muscle between bouts, thereby increasing total oxygen and nutrient delivery while promoting the removal of metabolites known to cause fatigue. PCr resynthesis is also highly dependent on local oxygen availability [24,25] which is severely diminished during a single bout of all-out effort high-intensity exercise. Given that the potential mechanisms thought to underpin exogenous ATP’s ergogenic benefit are proposed to manifest during periods of rest [9] and following exercise [6], increases in metabolite removal and PCr resynthesis may explain the reported benefits of exogenous ATP during repeated bout exercise that would otherwise not be observed during a single high-intensity all-out effort. It should be noted that increases in whole blood or plasma ATP concentrations following exogenous ATP supplementation are not generally observed, prompting uncertainty regarding its ergogenic potential. However, chronic oral ATP supplementation has been shown to increase portal vein ATP concentrations and nucleoside uptake by red blood cells (RBC’s) in rats, resulting in an increase in RBC ATP synthesis, despite no apparent elevation in plasma ATP levels [7]. Consistent with this, Purpura et al. [6] proposed that orally ingested ATP and its metabolites may work indirectly to increase ATP synthesis during high-intensity activities independently of increased plasma ATP levels. Additional research is needed regarding the potential mechanism underpinning the ergogenic benefits of exogenous ATP which were previously observed in the form of a disodium salt.

### Limitations

In the current study, participants were not familiarized with the 3MT prior to the first experimental trial. While a familiarization trial is commonly utilized prior to 3MT testing [26,27], differences in performance outcomes between familiarization and experimental trials are rarely reported; therefore, data on the familiarization effects of 3MT testing are lacking. Theoretically, the absence of a familiarization trial could have resulted in the utilization of a different pedaling strategy during completion of the second 3MT compared with the first that otherwise may have been established during familiarization. However, by randomizing and counterbalancing which treatment was provided during the first experimental trial, any differences in 3MT performance that were unrelated to the treatment should have been minimized. In consideration of this, an exploratory t-test between 3MT one and 3MT two (regardless of which treatment was provided first) revealed no significant differences in performance. Secondly, our study featured a heterogenous sample inclusive of men and women with different activity levels and varying degrees of previous training experience. Although consistent verbal encouragement was provided during the 3MT, and each participant appeared to provide maximal effort during each 3MT, participants with less training experience may have lacked a substantive perception of the physical and mental demands of a true maximal effort compared with their better trained counterparts. Training status was not evaluated in the current study; therefore, we were not able to characterize these differences. Nevertheless, since participants served as their own controls, we expect that any potential differences relating to training experience had a minimal impact on the findings. Thirdly, we did not control for diet or instruct participants to replicate their nutrition in the 24-h prior to each exercise trial. Although we provided an acute dose of the participants’ assigned supplements 30 min prior to the completion of each exercise trial, differences in participants’ caloric consumption or nutritional composition prior to each exercise trial may have impacted their 3MT outcomes. Lastly, this experimental trial was originally powered to detect changes in Go/No-Go accuracy based on data reported by Sun. et al. [28] as part of a separate research question aimed at examining the effects of PeakATP on cognitive performance following the 3MT. Nevertheless, the sample size of this investigation is equivalent to or exceeds that of prior investigations examining the ergogenic effects of disodium ATP supplementation on exercise performance. Therefore, if there were significant differences between the treatments regarding the 3MT performance parameters that were assessed in the current study, the sample size and study design utilized herein was sufficient to detect these differences.

## 5. Conclusions

In conclusion, to our knowledge, this study is the first to examine the effects of two weeks of 400 mg disodium ATP supplementation on performance during a 3MT. Our investigation found no effects of oral ATP supplementation on PP, TTP, WEP EP, and FI. While the 3MT shares fundamental characteristics with many of the previously tested anaerobic assessments, the 3MT is a continuous exercise with no rest periods. This may diminish the potential mechanistic effects of exogenous ATP and could explain the lack of benefit when compared with PLA. The 3MT also entails a much larger aerobic requirement than many of the previous investigations’ assessments, which may additionally explain the lack of agreement between our findings and previous findings. Future research may consider a comparative examination of ATP’s effects on repeated bout and single bout exhaustive exercises simultaneously to further characterize under which conditions ATP may effectively enhance performance.

## Figures and Tables

**Table 1 jfmk-08-00042-t001:** Descriptive statistics of the participants’ characteristics.

	All (*n* = 20)	Males (*n* = 10)	Females (*n* = 10)
Age (y)	22.3 ± 4.4	21.6 ± 1.6	23.0 ± 6.1
Height (cm)	169.9 ± 9.5	176.9 ± 6.9	163.0 ± 6.0
Weight (kg)	78.7 ± 14.6	85.5 ± 12.1	72.0 ± 14.1
Lean mass (kg)	58.4 ± 14.6	69.5 ± 11.2	47.4 ± 5.2
Body fat (%)	27.0 ± 9.5	20.7 ± 6.8	33.3 ± 7.6
Body water (kg)	42.7 ± 10.4	50.9 ± 8.3	34.6 ± 3.9

Data from the 20 participants (10 male and 10 female) are presented as means and standard deviations (SD). Body fat is expressed as a percentage value. cm = centimeters and kg = kilograms.

**Table 2 jfmk-08-00042-t002:** Between treatment differences in 3MT variables.

Variable	Treatment	Means (SD)			Mean Diff (SD)			Hedge’s *g*	Effect Size	*p*-Value
PP (W)	PeakATP	507	±	327	−8.30	±	115.13	0.023	Trivial	0.751
	PLA	499	±	331						
TPP (s)	PeakATP	3.33	±	4.43	0.71	±	6.56	−0.211	Small	0.636
	PLA	4.04	±	4.50						
WEP (kJ)	PeakATP	10.3	±	6.1	−0.03	±	1.56	0.006	Trivial	0.931
	PLA	10.3	±	6.5						
EP (W)	PeakATP	127	±	55	1.19	±	15.13	−0.030	Trivial	0.730
	PLA	128	±	52						
FI (%)	PeakATP	69	±	13	0.07	±	0.07	0.130	Trivial	0.424
	PLA	68	±	13						

Values are shown as the mean of each treatment ± the standard deviation. The mean difference is expressed as an absolute change. *p*-value = paired t-test; PP = peak power; TPP = time to peak power; WEP = work above end power; EP = end power; FI = fatigue index; W = watts; s = seconds; and kJ = kilojoules.

## Data Availability

The data presented in this study are available upon request from the corresponding author.

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
