# Peer review of "The Effects of Two Weeks of Oral PeakATP® Supplementation on Performance during a Three-Minute All out Test"

_jfmk, 2023, doi:10.3390/jfmk8020042_

Round 1

Reviewer 1 Report

L21: Only the last line of the abstract refers to the supplement as disodium ATP.  Did the previous work that is being referenced provide the patented PeakATP as well?  If so, it is recommended to refer to it as PeakATP for consistency in the abstract. 

L84: Please describe how diet was controlled prior to each trial.  Factors such as hours fasted, caffeine intake, duplicating a 24 hour recall should be considered.

L100: A power-analysis is mentioned in the discussion, therefore it is recommended to include information on power analysis to determine the number of participants needed in this section.

L104: The PAR-Q+ and MHQ acronyms have already been defined.

L136: The PPO acronym has already been defined.

L131: Throughout the manuscript, O2 and CO2 should be written using subscript.

L140:  Please provide a reference to previous work using the 3MT for which your protocol is based upon.

L158: PLA acronym has already been defined.

L170: Asking participants to consume a supplement 30 minutes prior to breakfast on an empty stomach can be difficult as it may require participants to wake up earlier than they would normally.  Please report compliance data regarding intake and timing.

L214: For consistency, it is recommended to change ATP to PeakATP in Tables.

L225: replace ; with ,

L228: this is the first time in the manuscript that null effects of PeakATP have been reported.  It should be noted in the introduction that there are some contradictory study outcomes as the introduction made it seem like all the evidence was currently positive.

L229: It should be noted that Jordan et al (2004) showed no effect of ATP supplementation (both 150 and 225mg) on blood ATP levels, which provides a major limitation on speculation on the mechanism by which supplementation may work.

L234 & 244: The Rathmacher study is said to have shown attenuations in fatigue in the introduction.  Please add clarity to these somewhat contradictory statements.

L248: It might be worth mentioning that there have been methodological issues brought up regarding the Wilson data (e.g., https://nutritionandmetabolism.biomedcentral.com/articles/10.1186/s12986-017-0201-7)

L255: Is the use of NO2 correct here? Or should this be nitric oxide (NO)?

L273: Pcr should be PCr 

L309: investigation's should be investigations'

Author Response

Thank you for your time and consideration of our article. Responses to your points have been addressed to the best of our abilities and are included in the attached word document.

Reviewer 2 Report

I read this twice and I think it is excellent.  I am wish I had more constructive feedback to give. 

Author Response

(The authors gave the same response as above.)

Reviewer 3 Report

The 3MT is an unusual exercise test but it seems the participants were not familiarized for the 3MT. Other studies (e.g. DOI: 10.1249/mss.0b013e31802dd3e6 and DOI: 10.1249/MSS.0000000000002395) have use full familiarization trials. Please provide a comment and discuss the consequences of the absence of a full 3MT familiarization trial in the study. There is some mention in L281 “It is feasible that this could have led to differences in performances between the two treatments depending on the order in which each treatment was provided.” but still it is unclear how the absence of a full familiarization would affect testing in one condition different than in the other.

Although a treatment order effects may have been mitigated, as suggested, the authors should provide the data that there was no order effect. This information will

L15. How long was the wash-out. Please provide that information in the abstract.

L27. “ATP is a part of most daily diets, with relatively high concentrations occurring naturally in meat, fish, and nuts.” Please provide a reference with composition analysis for ATP in those food sources. However, I do think that the statement is incorrect so please reconsider.

L45. I suggest to clarify that the observations from [12,13] were not in-vivo.

L55. The study by Rathmacher et al [15] did not show enhanced muscle excitability. Please check and revise.

L55. “declines in performance during later bouts of repeated exercise”. Please be more specific what kind of performance and what kind of exercise.

L73 and L13 (abstract) and L159 are not consistent on the experimental design. Please revise.

L132. “during the last completed stage of the test”. Not sure whether the test had any stages and increments were 1 W every 2 seconds. Please clarify.

L134. Please provide the equation used for age-predicted maximum heart rate.

L135. “a plateau in oxygen consumption despite an increase in exercise intensity”. Intensity increased every 2 seconds. Was it a plateau for 2 seconds? Please clarify.

L172. Please report on the side effects.

L174. Please provide the information on supplementation/placebo that was not consumed.

L187. For example, an effect size of 0.6 would be considered small and moderate. Please revise so that each effect size value can only be described with one descriptor.

L193. Please provide a reference for equation 1.

Table 2. G not capital letter.

Table 2. I suggest to provide EP values without decimals. Similar for FI(%).

L236. “It should be noted that ATP was not give the morning of the exercise bout.” Was this in the Rathmacher et al study. Please clarify.

L244. “in later bouts of a repeated”. Bouts and repeated seems saying it twice. Please revise.

L245. “fatigue of the leg”. Clarify which muscle group.

L252. Please provide references that some tests such from tissue ischaemia. How was that measured in those studies.

L259. “PCr synthesis within the mitochon dria. “ PCr synthesis does not happen in the mitochondria. Please revise.

L273. Change “Pcr” to “PCr”.

L304. Change “no significant effects” to “no effects”.

Author Response

(The authors gave the same response as above.)

Round 2

Reviewer 3 Report

Thanks for addressing my comments and suggestions.